# The Controversial Nature of Some Non-Starter Lactic Acid Bacteria Actively Participating in Cheese Ripening

**DOI:** 10.3390/biotech12040063

**Published:** 2023-11-09

**Authors:** Svetoslav G. Dimov

**Affiliations:** Department of Genetics, Faculty of Biology, Sofia University “St. Kliment Ohridski”, 1504 Sofia, Bulgaria; svetoslav@biofac.uni-sofia.bg; Tel.: +359-2-8167342

**Keywords:** non-starter lactic acid bacteria (NSLAB), *Lactococcus garvieae*, *Streptococcus uberis*, *Streptococcus parauberis*, *Weissella hellenica*, *Weissella confusa*, *Weissella paramesenteroides*, *Weissella cibaria*, *Mammalicoccus sciuri*

## Abstract

This mini review deals with some controversial non-starter lactic acid bacteria (NSLAB) species known to be both human and animal pathogens but also health-promoting and probiotic. The focus is on *Lactococcus garvieae*, two *Streptococcus* species (*S. uberis* and *S. parauberis*), four *Weissella* species (*W. hellenica*, *W. confusa*, *W. paramesenteroides,* and *W. cibaria*), and *Mammalicoccus sciuri,* which worldwide, are often found within the microbiotas of different kinds of cheese, mainly traditional artisanal cheeses made from raw milk and/or relying on environmental bacteria for their ripening. Based on literature data, the virulence and health-promoting effects of these bacteria are examined, and some of the mechanisms of these actions are reviewed. Additionally, their possible roles in cheese ripening are also discussed. The analysis of the literature data available so far showed that, in general, the pathogenic and the beneficial strains, despite belonging to the same species, show somewhat different genetic constitutions. Yet, when the safety of a given strain is assessed, genomic analysis on its own is not enough, and a polyphasic approach including additional physiological and functional tests is needed.

## 1. Introduction

Recently, customers worldwide have shown increased interest in consuming fermented dairy foods given their perceived properties as functional foods and their potential health benefits. In this regard, special attention was given to traditional, artisanally produced kinds of cheese, which are highly priced and which, in many cases, are prepared from raw milk and/or they rely on environmental microbiota for their ripening. Commonly, this environmental microbiota comes from the ambient environment (waters, pastures, air, etc.), but it also comes from human and animal external (skin, furring) and internal environments (gastrointestinal tract (GIT), mammary glands, etc.). Environmental bacteria can be commensal or pathogenic in nature. The bacteria that occur in milk, which is a very nutritionally rich environment, can adapt to this new environment and assimilate the milk sugars, proteins, and fats. These adaptations are in two main directions, the loss of some virulence determinants and the acquisition of genetic changes, allowing better assimilation of the milk’s nutrients. Often, by becoming part of the dominant autochthonous microbiota, these “new dairy bacteria” play an essential role in cheese ripening by contributing to specific organoleptic and rheological properties [1,2,3].

Typical examples of controversial NSLAB species, which actively participate in cheese ripening, are the members of the genus *Enterococcus*, which also could be opportunistic human and animal pathogens or probiotics, and have been investigated in both of these aspects for decades [4]. However, in the last decade, with the advent of the next-generation sequencing (NGS) techniques used for metagenomic studies of different kinds of cheese, several ubiquitous newcomers were revealed within the group of controversial NSLAB. Some of the most controversial omnipresent NSLAB belong to the genera *Lactococcus*, *Streptococcus*, *Weissella* and *Mammalicoccus* (Table 1).

## 2. *Lactococcus garvieae*

The *Lactoccus* genus was separated from the genus *Streptococcus* in 1985 as result of thorough DNA-based analysis such as nucleic acid hybridization studies and immunological relationships of superoxide dismutase [37], together with the early reported species as *Streptococcus garvieae* [38]. *L. garvieae* is a species with a pronounced dualistic nature, participating in the ripening of many cheeses worldwide but also being known as a pathogen. This species is mainly known as the causative agent of fish lactococcosis associated with hyperacute and hemorrhagic septicemia, leading to substantial economic losses [39]. The species is generally considered safe for humans and farm animals; nonetheless, it has been associated with bovine mastitis [40]. Rare cases of endocarditis in old and immunocompromised persons were also reported [41], as well as affecting patients with prosthetic valves [42] (Table 2). On the other hand, *L. garvieae* has been reported to be present in different environmental niches, including plant sprouts [43], as well as in different fermented foods such as fermented sausages [44], but mainly in fermented dairy products, including many types of cheese worldwide.

One of the earliest reports of *L. garvieae* in cheese dates from 2001, when it was found to be part of the microbiota of some traditionally prepared mozzarellas [5,6]. Later, it was reported worldwide to be found in many kinds of cheese prepared mainly from raw milk and/or without the addition of starter cultures, such as the Italian Toma Piemontese cheese [7], the Spanish Casín cheese [8] and “Torta del Casar” cheese [9], the Slovakian May bryndza cheese [10], the Azorean Pico cheese [11], some traditional Montenegrin brine cheeses [12], some Bulgarian and Turkish “Skin bag” (Tulum) cheeses [13,14], and the Bulgarian “Green” [15] and Krokmach [16] cheeses (Table 1).

It has been reported that when present within the dominant microflora, dairy *L. garvieae* strains positively contribute to cheese ripening and palatability [68], and they are also partially responsible for the typical sensorial characteristics of the final product [69]. It has been proven that dairy-related strains are lactose fermenters, despite the relatively slow acidification rate [69] (Table 3). Their presence does not affect the main physicochemical properties such as humidity, water activity, pH, texture, or color while contributing positively to the aroma with the production of methyl-branched acids and reducing the oxidation compounds originating from the β-oxidation of fatty acids present within the milk [70].

Inhibitory activity against pathogens and spoiling agents has also been reported for some dairy *L. garvieae* strains. Some of the main manifestations of this property are the documented inhibition of *Listeria monocytogenes* [9] and *Staphylococcus aureus* [71]. This bacteriostatic effect could be due to nutritional competition or hydrogen peroxide production [72]. However, *lactococci* are known bacteriocin producers, and dairy-related members of the genus are not an exception. Some examples are the broad-spectrum bacteriocins garvicin KS with inhibitory activity against *Bacillus*, *Listeria*, *Enterococcus,* and *Staphylococcus* [73] and garviecin L1-5 with inhibitory activity against *Clostridium*, *Enterococcus*, *Lactococcus,* and *Listeria* [74] (Table 3). These inhibitory and/or bacteriostatic properties are currently heavily exploited, and to control *Listeria* growth, some authors propose the addition of selected *L. garvieae* strains as NSLAB within starter cultures [9] and even their inclusion within edible cheese coatings [97].

There is much scientific proof that dairy-derived *L. garvieae* strains show different genetic constitutions from the pathogenic ones. First, they can grow on milk because of their ability to assimilate lactose. Fortina et al. [7] report that dairy isolates possess the genes necessary for lactose catabolism, while these genes are absent in the fish pathogens. Furthermore, these genes are located on the bacterial chromosome in contrast to the cheese “big classic” *L. lactis* [98]. These observations are further confirmed by the study of Foschino et al., who found that *L. garvieae* from the two ecological niches are genetically divergent, even with the limitations of the DNA fingerprint techniques which were widely used at this time [99]. Additionally, dairy-derived strains lack some of the pathogenicity phenotypes: they are non-agglutinating [43], they do not produce hemolysins and gelatinase, and many of the dairy strains lack the tetM and tetS genes encoding tetracyclines resistances. [69] All these findings suggest that they have low virulence and pathogenicity profiles [12].

Taking these issues into account, *L. garvieae* should be considered an essential and promising NSLAB that contributes positively to the ripening process and to the quality of the product. Nonetheless, to be applied as an additive to starter cultures, because of the controversial nature of the species, a thorough study of each strain should be conducted to guarantee its safety, for example, through whole-genome sequencing combined with phenotype characteristics [12].

## 3. *Streptococcus uberis* and *Streptococcus parauberis*

Half a century ago, *S. uberis* was reported to be the causative agent of clinical and subclinical cases of bovine mastitis [45]. A decade later, based on some phenotypic characteristics, *S. parauberis*, which was also documented as a bovine mastitis causative agent, was separated from *S. uberis* as a different species [100]. The new species also turned out to be a fish pathogen [47], while *S. uberis* was only detected in water environments and fishes without being associated with pathogenesis [101]. It has been documented that these species possess good environmental survival capabilities [102], which can explain why they are responsible for a significant proportion of clinical mastitis cases [103]. *S. uberis* has been occasionally associated with human infection; however, there is scientific evidence that in these cases, it has probably been misidentified [46]. In recent years, in the scientific literature, rare cases of infections in humans caused by *S. parauberis* have been reported [48,49] (Table 2). Although, in both cases, traumatism was involved, and human biological barriers were not passed through in a natural way. Both species have been shown to be present in different ecological niches in dairy farms, such as wastewater disposal sites, raw milk, udder, cow skin, grass, and soil [104].

Since both species are widely spread in the environment, and because of their ability to infect the cattle mammary glands, it is not surprising to find them in milk and fermented dairy products prepared from raw milk (Table 1). *S. uberis* has been detected for the first time among the dominant microbiota of a mozzarella cheese [6], while *S. parauberis* was reported as a dominant species for the Spanish blue-veined Cabrales cheese [18]. In combination or separately, both species have been observed in high amounts in many kinds of cheese worldwide. Some examples include the traditional Spanish Casín cheese [8], the Iranian Lighvan and Koozeh cheeses [19], some Slovenian raw milk cheeses [20], the Slovakian May bryndza cheese [10], the Italian Casizolu [17], Giuncata and Caciotta Leccese [21] cheeses, the Turkish Tulum cheese [14], and the Bulgarian Mehovo sirene cheese [13].

The observation of high amounts of *S. uberis* and *S. parauberis* within the cheese microbiotas means that they play a role in the ripening process (Table 3). It was reported that *S. uberis* produces an extracellular protein named streptokinase which activates the plasminogen to active plasmin, which in turn, results in plasmin-induced proteolysis of the milk proteins [75]. Initially, this mechanism evolved for the development of mastitis; however, it also contributes to the ripening of the cheeses. The same mechanism of *S. parauberis* was observed and studied during the ripening process of the Azerbaijani Lighvan cheese [77].

*S. thermophilus* is known to contribute significantly to flavor development [105], so it is logical to expect that in the ripening process, other members of the genus should play, to some extent, the same role. Indeed, Yang et al. [78] report a positive correlation between some of the organoleptic properties of several cheese samples and the high content of *S. parauberis* within their microbiota. These authors explain their observation with the findings that some *S. parauberis* strains are capable of producing enzymes needed to produce linear alkanes and alcohols. 

In contrast to *L. garvieae* isolates which split into a pathogenic and dairy lineage, not surprisingly, no such observations have been detected for the *S. uberis* and *S. parauberis* isolates since they originate from environmentally infected cattle. Still, in addition to their participation in the cheese ripening process and the development of palatability, because of their ability to inhibit the growth of some other pathogens and spoiling agents, some isolates have additional beneficial effects on the final product (Table 3). Tulini et al. report the isolation of bacteriocin-producing *S. uberis* strains from Brazilian cheese inhibiting the growth of *Carnobacterium maltaromaticum*, *Latilactobacillus sakei*, and *Listeria monocytogenes* [76]. Antagonistic activity of *S. uberis* was also reported for several cheese isolates from Serbia, and the authors report that these isolates are also susceptible to antibiotics [106].

## 4. The Genus *Weissella*

Based on a comparative analysis of 16S rRNA genes, the *Weissella* genus was separated from the *Leuconostoc* genus in 1993, with *W. hellenica* as a novel species isolated from a type of Greek sausage [107]. Soon after, it became apparent that the genus possesses a controversial nature, comprising species with clear pathogenic potential and species with strong probiotic properties and potential for the food industry. Unfortunately, some species of the genus comprise strains with beneficial properties, but others have been proven to possess pathogenic properties [82]. 

Among *Weissella* species, mainly *W. hellenica*, *W. confusa*, *W. cibaria*, and *W. paramesenteroides* were reported to participate in the fermentation of dairy products [21]. Until now, there were no scientific reports on the association of *W. hellenica* and *W. paramesenteroides* with clinical cases or infections in humans or animals. In contrast, *W. confusa* is definitely a species with a dualistic nature—some isolates have been reported as pathogens while others have been reported as probiotics (Table 2 and Table 3). In addition to being found within the gastrointestinal tract of healthy humans [108], *W. confusa* has been reported to cause bacteremia [50] and endocarditis [51] in humans and even deadly infections in primates [52]. On the other hand, many strains of the same species possess different strong probiotic properties, including the antibacterial activity against *E.coli* of a strain isolated from kimchi [23,89,109,110]. *W. cibaria* was first considered as a human and animal commensal species which can be isolated from feces, saliva, and vaginal mucous; the species also emerged as an opportunistic pathogen associated with human blood and lung swab bacteremias, as well as being isolated from human urine [53]. It has also been linked to otitis in dogs [54]. Similarly to *W. confusa*, for many *W. cibaria* isolates, probiotic properties have been documented [88,111].

The different *Weissella* species have been reported to be part of the microbiotas of many kinds of cheese worldwide, mainly artisanal and/or those prepared from raw milk (Table 1). *W. hellenica* was reported to be found in Danish raw milk cheeses [22], Croatian cheese [20], several Brazilian artisanal cheeses [23], and traditional Italian mozzarella cheese [24]. Some examples of cheeses in which *W. paramesenteroides* are present within their microbiota include a type of a Mexican ripened cheese [28], some traditional French cheeses [112], the Columbian double cream cheese [29], the Greek hard cheese Manura [30], and the traditional Turkish Sepet cheese [25]. *W. confusa* was also reported to be present in the latter [25]. It was also found within the microbiotas of Kazak cheese [26] and a specific kind of Indonesian cheese [27]. Similarly to the other three species, *W. cibaria* has been reported to be part of the microbiotas of different cheeses around the globe—within the West African Tchoukou cheese [31] and within a cheese from the Western Himalayas [32]. Moreover, because of their probiotic properties, some *W. cibaria* are often added as adjunct cultures [81].

The role of *Weissella* species in cheese ripening is to a great extent linked to their beneficial and health-promoting effects due to the synthesis of exopolysaccharides (EPS) or the inhibition of pathogens [82,83,113] (Table 3). By synthesizing EPS [32,79], they contribute to the rheological properties. On the other hand, due to their ability to produce lactic acid and other low molecular weight acids by assimilating lactose and galactose, they not only inhibit the growth of some potential pathogens and soiling agents, but they also contribute to the coagulation of milk proteins [23,80]. Another significant role of dairy *Weissella* isolates is related to lipolytic and proteolytic activities, which in turn contribute to the development of the aroma and flavor [81]. Many strains are reported to produce the volatile compound diacetyl resulting from the conversion of citrate to pyruvate and are related to the “buttery” aroma [23].

Many different *Weissella* spp. isolates have been proven to possess probiotic and health-promoting effects such as the production of EPS; they possess antioxidant activity, can transform prebiotics, and have antimicrobial activities due to the production of hydrogen peroxide, organic acids, and bacteriocins. For *W. cibaria*, *W. confusa*, and *W. paramesenteroides*, which can also be found in cheese and dairy environments, good survival capabilities within the gastrointestinal tract (GIT), alongside the ability to transform prebiotic fibers, have been reported [83].

Different *Weissella* isolates from different types of samples are among the most potent producers of different types of linear and branched EPS, such as glucans, dextrans, mannose, and glucose and galactose homo- and heteropolysaccharides. For many of them, beneficial biological probiotic and prebiotic properties such as antioxidant activity, antimicrobial activities, immunomodulatory activity, prebiotic potential, and stimulation of the growth of probiotic bacteria have been reported [82,84]. Cheese-derived *W. cibaria* and *W. confusa* isolates are also reported to be EPS producers [32,85]. Because of both the EPS’s health-beneficial effects and their attribution to the rheological properties of cheese, EPS-producing strains are often added as adjunct NSLAB cultures in cheese production [81].

One of the mechanisms of the antibacterial activity against pathogens of *Weissella* spp. is the production of bacteriocins [79,86,87]. Yet, the antimicrobial action against pathogens can also result from the synthesis of organic acids, EPS, or hydrogen peroxide [83,111]. Hydrogen peroxide production has been proven to have an oral health-promoting effect such as the inhibition of *Streptococcus mutans* and *Fusobacterium nucleatum*, which are causative agents of plaque formation and periodontitis [82,88]. A *W. confusa* isolate was reported to inhibit *Helicobacter pylori*’s growth and to block its binding to the stomach [89]. The antilisterial and antioxidant activities of a *W. cibaria* isolate were exploited with its addition as an adjunct NSLAB culture [113]. Furthermore, antifungal activities of food-isolated *Weissella* strains were discovered. A *W. paramesenteroides* strain was shown to inhibit food molds with the production of phenyllactic acid, 2-hydroxy-4-methylpentanoic acid, and other organic acids [90], while a *W. cibaria* sourdough isolate showed potent inhibitory activity against *Aspergillus niger*, *Penicillium roqueforti,* and *Endomyces fibuliger* with an uninvestigated mechanism [91].

Some additional health-promoting and beneficial effects have been identified in some *Weissella* spp. strains. For example, both antitumor and chemo-preventive effects [92] and anti-obesity effects [93] have been reported. Immunomodulating, anti-inflammatory, and antiviral activity have also been observed [82].

Because of the many probiotic and health-promoting effects, *Weissella* spp. are of great interest to the pharmaceutical and food industries. Yet, because of the controversial, dualistic nature of the representatives of the genus, one should take great caution before attributing a “generally recognized as safe” (GRAS) status to a *Weissella* isolate. Without any doubt, each promising isolate should be investigated separately. It can be carried out via whole genome sequencing and bioinformatic analysis for the presence of genes encoding probiotic determinants and genes encoding virulence factors. Though, in silico analysis is not enough to assess the virulence potential of a given isolate because some genetic determinants for virulence factors are intrinsic to the genus, as is the case for many LAB of other genera with a GRAS status, while at the same time, some other genetic determinants could contribute to the probiotic potential [82]. So, to characterize newly isolated *Weissella* strains, a polyphasic approach comprising both genomic analyses and physiological and functional tests would give the most accurate results [114].

The presence of genes encoding haemolysins and haemolysin-like proteins appears to be ubiquitous in many LAB [114] and can often be revealed with in silico analyses of *Weissella* genome sequences [82]. For this reason, it is largely believed that they should not be regarded as an exclusion factor for a probiotic isolate [82,114].

Another potential trait of concern is the presence of antibiotic resistance (AR) genes. Yet, the resistance to some commonly used classes of antibiotics, such as glycopeptides (vancomycin), aminoglycosides (gentamycin, kanamycin), and sulphonamides, is in many cases intrinsic to many LAB, including several *Weissella* spp. [114]. In general, if the antibiotic resistance genetic determinants are not located on mobile genetic elements or plasmids, they cannot be assessed as virulence factors because they are considered intrinsic [82]. So, in the case of AR, only phenotypic characterization is insufficient, and in-depth genomic analysis is needed to assess the virulence potential of a given *Weissella* strain.

Adhesins are another factor that could raise a concern. In pathogenic bacteria, they play an essential role in the colonization and interaction with the host [115]; yet, the same proteins also contribute to the colonization of the health-beneficial bacteria and block the adhesion of pathogens with a concurrence mechanism, as is the case of a probiotic *W. cibaria* isolate [116]. A significant role in the adhesion of the probiotic bacteria within the GIT is played by the mucus-binding proteins, so the presence of genetic determinants is considered a beneficial trait, as was reported for another *W. cibaria* cheese isolate [32].

## 5. *Mammalicoccus sciuri* (Formerly Known as *Staphylococcus sciuri*)

*Mammalicoccus sciuri* is a member of a group of bacteria formerly known as coagulase-negative staphylococci (CNS) before the taxonomic reclassification of some species of the genus *Stpahylococcus*. The CNS are a group of bacteria found among the predominant species in many fermented foods worldwide [95]. *M. sciuri* was first identified as *Staphylococcus sciuri* in 1976 as a new species of the so-called group III staphylococci, which were reported to be human and animal skin commensals [117]. It was reclassified in 2020 as a member of the new genus *Mammalicoccus* of the *Staphylococcaceae* family [118]. Within time, some *M. sciuri* strains were reported to possess strong pathogenic potential for humans and animals. In humans, it has been reported to be a causative agent of wound infections [55], urinary tract infections [56], endocarditis [57], sepsis in adults [59] as well as neonatal sepsis [58], endophtalmitis [60], peritonitis [61], and plevric inflammatory disease [62].

*M. sciuri* strains have been isolated mainly from warm-blooded animals, comprising farm animals, pets, and wild animals. Often, this species is found in a large variety of healthy farm animals such as pigs, poultry, sheep, goats, and horses [63], but also in a broad range of wild animals—rodents, carnivores, monkeys, and even cetaceans and marsupials [119]. However, potentially pathogenic strains are often recovered from farm animals such as pigs, cows, and broilers [120], and it is not surprising that members of this species are causative agents of mastitis in dairy cattle (cows and goats) [63,64], as well as severe epidermitis in piglets [65]. The species was also discovered in goats suffering from ovine rinderpest [66]. It has also been associated with fatal infections in pets (dogs and cats), causing acute respiratory distress syndrome [67] (Table 2).

Different CNS species have been identified as part of the dominant and subdominant microflora of many kinds of traditional cheeses [33,34] (Table 1). *M. sciuri*, despite being mainly associated with the ripening of fermented meat products such as cured meats and sausages [95,96,121], has also been found in French smear cheeses [33], German cheeses [34], Brazilian cheeses [35], and the traditional Middle East Surk cheese [36] (Table 1).

Even though it is not especially studied, the role of *M. sciuri* in the ripening of fermented foods could not be very different from that of other CNS. It has been reported that food-derived staphylococci contribute mainly to organoleptic properties (Table 3). This function is achieved thanks to the catabolism of carbohydrates and amino acids, but also the synthesis of esters. Small flavor compound molecules are also produced by some aspects of their proteolytic and lipolytic activities [95,96]. A correlation between the smell and the presence of *M. sciuri* has been investigated, and it was found that this species, in combination with some yeasts, is responsible for the olfactory characteristics of some green cheeses [94].

Although there are some sporadic reports of the isolation of *M. sciuri* strains with probiotic activity [122], in general, because of the species’ relatively strong pathogenic potential, as well as the lack of isolates with attributed GRAS status in the United States or QPS status in the European Union, the question of the health-promoting effects should be considered with great caution.

There are no comparative studies on the genetic lineages and constitution of pathogenic *M. sciuri* isolates and those derived from fermented foods, while similar investigations on other CNS are relatively scarce. One of the main concerns of using these bacteria as adjunct cultures is that they usually carry genetic determinants for virulence factors. An example is the presence of genes encoding hemolysins. Nevertheless, there are reports that the presence of such factors does not always imply a hemolytic phenotype, and their presence in food-derived CNS is generally sporadic [96]. Another concern is their ability to produce biogenic amines such as cadaverine, putrescine, histamine, and tyramine, which could cause food poisoning. However, comparative genomic analysis showed that they usually lack the necessary genes. These findings are greatly supported by the fact that until now, cases of food poisoning due to CNS were never reported [95].

The presence of AR genes could be another indicator of pathogenicity. Within the group of CNS, one of the most predominant AR is the methicillin resistance, encoded by the *mecA* gene. Interestingly, although the presence has been reported in many *M. sciuri* isolates, it is usually not sufficient to confer resistance, except if other regulators *mec*-genes are also present as part of a mobile genetic element known as staphylococcal cassette chromosome (SCCmec) [123]. So, similarly to the other dualistic NSLAB discussed already, the pathogenic potential of the AR genes depends on whether they are located in mobile elements, while the intrinsic AR represent low risk [96]. Additional hazard comes from the fact that *M. sciuri* possesses a vast range of habitats, including wild animals and environments, which could serve as a reservoir for pathogenicity determinants, which in turn could be passed to dairy strains via horizontal genetic transfer [120].

## 6. Conclusions

Taking into account the considerations above, based on genomic, functional, and physiological analyses, several conclusions for the controversial NSLAB of the genera *Lactococcus*, *Streptococcus*, *Weissella,* and *Mammalicoccus* could be made. First, it is scientifically proven that they contribute to the ripening of cheeses by influencing the organoleptic and rheological properties. Second, the food-related strains usually differ in their genetic constitution and phenotypic characteristics from the pathogenic strains. Third, many food- and dairy-related strains possess probiotic and health-promoting properties, giving characteristics of functional food to the products they ferment. Finally, to evaluate the safety of each isolate of these controversial genera, a polyphasic approach should be undertaken, combining genomic analyses, physiological, and functional studies. These analyses, especially those concerning whole-genome sequencing and the comparison of the sequences of health-promoting and pathogenic isolates, could be a starting point in the future for new taxonomic speciation for some of the strains of controversial species.

## Figures and Tables

**Table 1 biotech-12-00063-t001:** Some examples of controversial NSLAB participating in cheese ripening.

Genus	Species	Some Examples of Cheeses	References
*Lactococcus*	*L. garvieae*	Italian mozzarella cheeses Italian Toma Piemontese cheese Spanish Casín cheese Spanish “Torta del Casar” cheese Slovakian May bryndza cheese Azorean Pico cheese Montenegrian brine cheeses Bulgarian and Turkish Tulum cheeses Bulgarian “Green” cheese Bulgarian Krokmach cheese	[5,6] [7] [8] [9] [10] [11] [12] [13,14] [15] [16]
*Streptococcus*	*S. uberis*	Italian Mozzarella cheese Spanish Casín cheese Italian Casizolu cheese	[6] [8] [17]
	*S. parauberis*	Spanish Cabrales cheese Spanish Casín cheese Iranian Lighvan and Koozeh cheese Slovenian raw milk cheeses Slovakian May bryndza cheese Italian Casizolu cheese Italian Giuncata cheese Italian Caciotta Leccese cheese Bulgarian and Turkish Tulum cheeses	[18] [8] [19] [20] [10] [17] [21] [21] [13,14]
*Weissella*	*W. hellenica*	Danish raw milk cheeses a type of Croatian cheese Brazilian artisanal cheeses Italian Mozzarella cheese	[22] [20] [23] [24]
	*W. confusa*	Turkish Sepet cheese a type of Kazak cheese a type of Indonesian cheese	[25] [26] [27]
	*W. paramesenteroides*	a type of Mexican ripened cheese some traditional French cheeses Columbian double cream cheese Greek Manura cheese Turkish Sepet cheese	[28] [29] [30] [25] [26]
	*W. cibaria*	Afrikan Tchoukou cheese Western Himalayan cheese	[31] [32]
*Mammalicoccus*	*M. sciuri*	French smear cheeses some German cheeses some Brazilian cheeses Middle East Surk cheese	[33] [34] [35] [36]

**Table 2 biotech-12-00063-t002:** Examples of pathogenicity of the bacteria investigated in this study.

Species	Pathogenicity	References
*L. garvieae*	fish lactococcosis bovine mastitis endocarditis in immunocompromised and old persons	[39] [40] [41]
	patients with prosthetic valves	[42]
*S. uberis*	bovine mastitis occasional human infections	[45] [46]
*S. parauberis*	bovine mastitis fish pathogen rare cases of infection in humans	[45] [47] [48,49]
*W. hellenica*	no records	
*W. confusa*	bacteremia endocarditis deadly infections in primates	[50] [51] [52]
*W. paramesenteroides*	no records	
*W. cibaria*	bacteremias in humans otitis in dogs	[53] [54]
*M. sciuri*	human wound infections urinary tract infections endocarditis in humans sepsis in humans endophtalmitis in humans peroitonitis in humans plevric inflammatory disease in humans mastitis in cows and goats epidermitis in piglets presence in ovine rinderpest suffering animals respiratory distress syndrome in cats and dogs	[55] [56] [57] [58,59] [60] [61] [62] [63,64] [65] [66] [67]

**Table 3 biotech-12-00063-t003:** Contribution to the ripening and health-promoting effects of the NSLAB reviewed in this study.

Species	Contribution to the Ripening	References	Health-Promoting and Probiotic Effects	References
*L. garvieae*	palatability sensorial characteristics lactose fermentation aroma	[68] [69] [69] [70]	inhibition of pathogens	[9,71,72,73,74]
*S. uberis*	streptokinase induced proteolysis	[75]	inhibition of pathogens	[54,76]
*S. parauberis*	streptokinase induced proteolysis organoleptic properties	[77] [78]		
*Weissella* spp.	contribution to the rheological properties by EPS production	[32,79]	synthesis of EPS bacteriocins production hydrogen peroxide production inhibition of *H. pylori* antifungal activities chemopreventive effects anti-obesity effects antiviral activity	[32,79,80,81,82,83,84,85] [83,86,87] [82,88] [89] [90,91] [92] [93] [82]
	coagulation of the milk proteins	[53,80]		
	organoleptic properties	[23,81]		
*M. sciuri*	organoleptic properties	[94,95,96]	no definitive data	

## Data Availability

Not applicable.

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
