# Peer review of "The Controversial Nature of Some Non-Starter Lactic Acid Bacteria Actively Participating in Cheese Ripening"

_biotech, 2023, doi:10.3390/biotech12040063_

Round 1

Reviewer 1 Report

Comments and Suggestions for Authors

The study entitled “The ambiguous nature of some non-starter lactic acid bacteria actively participating in cheese ripening” is well discussed and the topic is very interesting. I have pointed out improvements that could help increase the manuscript's quality.

1.      Yes, I agree that the mentioned lactic acid-producing bacterial species are widely involved in pathogenesis and possess health-beneficial effects. It indicates that the species needs further sub-classification. So, I encourage authors to suggest and identify the prospects to solve this issue.

2.      These bacterial species were not only involved in cheese ripening but also found in various traditionally fermented foods. For example, W. confuse is identified in Kimchi, which is a traditional Korean cuisine (DOI: 10.1016/j.lwt.2019.05.089; DOI: 10.3329/bjp.v14i3.41545, which are worth citing).

Please discuss whether a few of these bacteria were also found in the healthy gut system or not (Example: W. confusa is found in a healthy gut, DOI: 10.1016/j.ijfoodmicro.2011.01.015).

Author Response

Dear Reviewer I,

“The study entitled “The ambiguous nature of some non-starter lactic acid bacteria actively participating in cheese ripening” is well discussed and the topic is very interesting. I have pointed out improvements that could help increase the manuscript's quality.”

I am grateful for the high appreciation of this theoretical work!

“1.      Yes, I agree that the mentioned lactic acid-producing bacterial species are widely involved in pathogenesis and possess health-beneficial effects. It indicates that the species needs further sub-classification. So, I encourage authors to suggest and identify the prospects to solve this issue.”

A sentence was added at the end of the conclusions section.

“2.      These bacterial species were not only involved in cheese ripening but also found in various traditionally fermented foods. For example, W. confuse is identified in Kimchi, which is a traditional Korean cuisine (DOI: 10.1016/j.lwt.2019.05.089; DOI: 10.3329/bjp.v14i3.41545, which are worth citing).”

The suggested citations were included.

“Please discuss whether a few of these bacteria were also found in the healthy gut system or not (Example: W. confusa is found in a healthy gut, DOI: 10.1016/j.ijfoodmicro.2011.01.015).”

The suggested citation was included.

Reviewer 2 Report

Comments and Suggestions for Authors

The submission is a mini-review that focuses on Gram-positive cocci that are non-starter lactic acid bacteria (NSLAB) which occur as part of the complex microbiota particularly in artisanal cheeses made from raw milk.  NSLAB can contribute to fermented dairy product traits by influencing the organoleptic and aroma profiles which make regional products distinctive.  While NSLAB may also contribute to health benefits as probiotics, like many bacteria which are considered generally safe some NSLAB are opportunistic pathogens in compromised individuals or cause animal/fish/plant diseases.  There is a lot of valuable information summarized in this submission and there are few published articles covering the range of species described.  However, the author needs to consider editorial, phraseological and technological matters raised in the annotated pdf provided to assist in writing the next version of the text.

Editorial matters

1.       Title:  several terms are used in the text to imply that some of the species described are both valuable as NSLAB or probiotics but may also propose risks as opportunistic pathogens (or some isolates are known pathogens in, for example, fish).  The Abstract, and later in the text, requires a clear definition of what the term ‘ambiguous’ means, as this term means ‘uncertain’ or subject to multiple interpretations.  What the author is saying is that some of the species are GRAS, or there is insufficient documented data, but have pathogenic traits which may/may not exclude their use as adjunct cultures.  Consistency is needed in Table headings and text for clarity:  ambiguous, duality, controversial, ambivalent are all used in different places, so this is not at all clear currently.

2.       Numbering of references:  journal format requires checking, as sequential numbering does not occur in text and Table 1.

3.       Genus abbreviations:  Lactococcus is normally abbreviated to L., Streptococcus to S.:  please use conventional microbiology terminology.  Check all genus/species names and make sure these are in italics (some instances where this has not occurred are noted on the pdf, but not all).

4.       Introduction, lines 26-47:  this section is particularly badly written and needs review for word usage, sentence content/structure, using references to substantiate statements (most statements are obvious but useful to back up opinion with literature).

5.       Reference list:  review for journal format.

Technical matters

1.       Line 44:  mentions next-generation sequencing as an important aspect of taking NSLAB analysis forward.  However, many of the references cited did not use this and it is not clear what the current status of genome sequencing is for the species described.  This means that the actual identity of some isolates may, or may not, be accurate, particularly given the noted confusion of L. garvieae with enterococci in clinical settings.  A recent review (2019) is provided for information and use.  As the author concludes (line 106 and Conclusion) that genome sequencing and analysis of physiological/biochemical traits is required in the future to validate use of NSLAB as safe for use as adjuncts, providing guidance on the limitation of prior published works is important.  If there are few genomes published, from strains from a large diversity of environment niches, then state this clearly.  This is, indeed, stated in a later section for M. sciuri (lines 309-311).

2.       In the following sections, there is a brief history of the nomenclature of some of the genera and species, given name changes over the last 30 years with increasing numbers of genomes sequenced and phylogenetic relationships clarified.  It would be useful to add a similar brief history for L. garvieae for consistency and information, so the reader does not need to go back to source material to find out about the history of nomenclature.

3.       It would also be useful to indicate when the various species cause disease – L. garvieae has mainly been reported in compromised individuals or following eating diseased fish (see 2019 review), and also mentioned later for one of the other groups.

Overall

Consider resubmission with the editorial and technical matters addressed.

Comments on the Quality of English Language

Most of the manuscript text is understandable, although some of the sentences have unconventional phrasing.  The first part of the Introduction is poorly phrased, as noted in the report - almost as if another author composed this section.  

Author Response

Dear Reviewer II,

“The submission is a mini-review that focuses on Gram-positive cocci that are non-starter lactic acid bacteria (NSLAB) which occur as part of the complex microbiota particularly in artisanal cheeses made from raw milk.  NSLAB can contribute to fermented dairy product traits by influencing the organoleptic and aroma profiles which make regional products distinctive.  While NSLAB may also contribute to health benefits as probiotics, like many bacteria which are considered generally safe some NSLAB are opportunistic pathogens in compromised individuals or cause animal/fish/plant diseases.  There is a lot of valuable information summarized in this submission and there are few published articles covering the range of species described.  However, the author needs to consider editorial, phraseological and technological matters raised in the annotated pdf provided to assist in writing the next version of the text.”

I am especially grateful for the high evaluation of my work!

“Editorial matters

  1. Title:  several terms are used in the text to imply that some of the species described are both valuable as NSLAB or probiotics but may also propose risks as opportunistic pathogens (or some isolates are known pathogens in, for example, fish).  The Abstract, and later in the text, requires a clear definition of what the term ‘ambiguous’ means, as this term means ‘uncertain’ or subject to multiple interpretations.  What the author is saying is that some of the species are GRAS, or there is insufficient documented data, but have pathogenic traits which may/may not exclude their use as adjunct cultures.  Consistency is needed in Table headings and text for clarity:  ambiguous, duality, controversial, ambivalent are all used in different places, so this is not at all clear currently.”

The word “ambiguous” was replaced with “controversial.”

“2.       Numbering of references:  journal format requires checking, as sequential numbering does not occur in text and Table 1.”

The numbering was corrected.

“3.       Genus abbreviations:  Lactococcus is normally abbreviated to L., Streptococcus to S.:  please use conventional microbiology terminology.  Check all genus/species names and make sure these are in italics (some instances where this has not occurred are noted on the pdf, but not all).”

I am thankful for this remark and replaced “Lc.” with “L.” and “Str.” with “S.”.

  1. Introduction, lines 26-47:  this section is particularly badly written and needs review for word usage, sentence content/structure, using references to substantiate statements (most statements are obvious but useful to back up opinion with literature).

“5.       Reference list:  review for journal format.”

The reference list was generated using a reference manager using the MDPI style.

“Technical matters

  1. Line 44:  mentions next-generation sequencing as an important aspect of taking NSLAB analysis forward.  However, many of the references cited did not use this and it is not clear what the current status of genome sequencing is for the species described.  This means that the actual identity of some isolates may, or may not, be accurate, particularly given the noted confusion of L. garvieae with enterococci in clinical settings.  A recent review (2019) is provided for information and use.  As the author concludes (line 106 and Conclusion) that genome sequencing and analysis of physiological/biochemical traits is required in the future to validate use of NSLAB as safe for use as adjuncts, providing guidance on the limitation of prior published works is important.  If there are few genomes published, from strains from a large diversity of environment niches, then state this clearly.  This is, indeed, stated in a later section for M. sciuri (lines 309-311).
  2. In the following sections, there is a brief history of the nomenclature of some of the genera and species, given name changes over the last 30 years with increasing numbers of genomes sequenced and phylogenetic relationships clarified.  It would be useful to add a similar brief history for L. garvieae for consistency and information, so the reader does not need to go back to source material to find out about the history of nomenclature.
  3. It would also be useful to indicate when the various species cause disease – L. garvieae has mainly been reported in compromised individuals or following eating diseased fish (see 2019 review), and also mentioned later for one of the other groups.

Overall

Consider resubmission with the editorial and technical matters addressed.”

Almost all the technical matters mentioned as comments of the pdf file were addressed.

Round 2

Reviewer 1 Report

Comments and Suggestions for Authors

The manuscript entitled "The ambiguous nature of some non-starter lactic acid bacteria actively participating in cheese ripening" has been revised extensively and the present format is way better than the previous submission. I would be happy to accept this article in the current format for publication.